# Retrospective Single-Center Analysis of 5575 Spinal Surgeries for Complication Associations and Potential Future Use of Generated Data

**DOI:** 10.3390/jcm14020312

**Published:** 2025-01-07

**Authors:** Yoram Materlik, Volker Martin Tronnier, Matteo Mario Bonsanto

**Affiliations:** Department of Neurosurgery, University of Luebeck, 23562 Luebeck, Germanymatteo.bonsanto@uksh.de (M.M.B.)

**Keywords:** complications, spinal surgery, standardized data registration, logistic regression, risk stratification, artificial intelligence

## Abstract

**Background**: This study aims to retrospectively detect associations with postoperative complications in spinal surgeries during the hospitalization period using standardized, single-center data to validate a method for complication detection and discuss the potential future use of generated data. **Methods**: Data were generated in 2006–2019 from a standardized, weekly complications conference reviewing all neurosurgical operations at the University Hospital Luebeck. Paper-based data were recorded in a standardized manner during the conference and transferred with a time delay of one week into a proprietary complication register. A total of 5575 cases were grouped based on the diagnosis, surgical localization, approach, instrumentation, previous operations, surgery indication, age, ASA score, and pre-existing conditions. Retrospective analysis was performed using a logistic regression detecting complication associations. The results were compared to the literature validating the method of complication detection. **Results**: Mean cohort age: 58.83 years. Overall complication rate: 10.9%. Mortality rate: 0.25%. The statistically significant complication associations were age; an age of >60; the localization (cervical, thoracic); a cervical tumor or trauma diagnosis; lumbar degenerative conditions, tumor, trauma, or infection; a cervical hemi-/laminectomy and vertebral body replacement; a lumbar hemi-/laminectomy, posterior spondylodesis, and 360° fusion; lumbar instrumentation, with an ASA score of three and four; a ventral and combined/360° approach; a lumbar combined/360° revision; two, three and ≥four pre-existing conditions; hypertension; osteoporosis; arrhythmia; an oncological condition; kidney dysfunction; stroke; and thrombosis. **Conclusions**: Documenting risk profiles for spinal procedures is important in identifying postoperative complications. The available data provide a comprehensive overview within a single center for spinal surgeries. Standardized complication recording during an established complication conference in the clinical routine enables the detection of significant complications. It is desirable to standardize the registration of postoperative complications to facilitate comparability across different institutions. The results may contribute to national or international databases used for automated AI risk profiling.

## 1. Introduction

The 1999 Institute of Medicine (IOM) report “To Err is Human—Building a Safer Health System” brought attention to the substantial risks of medical errors and complications, prompting urgent calls for improved patient safety globally [1]. The findings indicated that medical errors and associated complications represented a substantial problem, at the time often overlooked in American healthcare, a conclusion applicable to other countries as well [2]. Over the past 24 years, advancements in peri- and postoperative complication management have included interdisciplinary communication, morbidity and mortality (M&M) conferences, targeted medical staff training, surgical simulator training, and error reporting systems [2]. Concurrently, recent technological advancements, particularly in artificial intelligence (AI), aim to create individual risk profiles for patients to enhance evidence-based decision-making in healthcare [3]. Hospital databases currently prioritize economic data management, foregrounding remuneration-related aspects and pushing medical queries back [4]. However, the accuracy of AI systems relies on the quality and relevance of available data. Standardized national and international patient registries with extensive case numbers and high data validity will be essential to harness the potential of AI for improving patient safety. Given the severe implications of medical procedure complications, careful monitoring and management are imperative. The objective of this retrospective study is to provide a comprehensive overview of postoperative complications associated with spinal surgeries, documented through a standardized methodology in a weekly single-center complications conference. The aim is to demonstrate a potential methodology for the standardized registration of postoperative complications in spinal surgeries, enabling outcome comparisons across medical centers. Through the analysis of a large spinal surgery cohort, we aim to add to the scientific database for future AI risk stratification efforts.

## 2. Materials and Methods

The design of a complication register was first initiated by the author in 1993 and implemented at the Department of Neurosurgery at the University of Heidelberg [5]. In 2005, the author introduced a corresponding register at the Department of Neurosurgery at the University of Lübeck. This single-center, retrospective study analyzed 5575 spinal surgeries in the Department of Neurosurgery, University Hospital Schleswig-Holstein, Campus Luebeck, from 1 January 2006 to 31 January 2019. Inclusion was limited to spinal operations following Bonsanto et al.’s definition [5]. The institutional review board of the University of Luebeck approved the study prior to it beginning (ref. no. 16-226), and patient data were pseudonymized with no additional patient interaction or external findings requested.

### 2.1. Data Generation

All surgical procedures performed were documented in a proprietary operation database (FileMaker Pro 17 Advanced 17.0.1.143, FileMaker Inc., Santa Clara, CA, USA) serving as the clinic’s internal operation register and interface for scientific data evaluation from 2006 onward. Overall, at the time of the study, this reference operation database included 19,444 treatments. From 2006 to 2007, surgical procedures were recorded using the C-KIS^®^ operating documentation program (c.a.r.u.s. HMS GmbH, Hamburg, Germany). The system transitioned to ORBIS^®^ (Agfa HealthCare GmbH, Bonn, Germany) from 1 January 2008 (Figure 1). Base data were manually sourced by the neurosurgical secretariat from the hospital information system (HIS), including the name, first name, date of birth, gender, age, operation date, operation duration, operating room, surgeon, and assistant. For diagnosis and treatment, standardized text formats were used. As part of this study, patient-specific ICD (International Statistical Classification of Diseases and Related Health Problems) and OPS (Operation and Procedure Keys) classifications documented in the HIS were imported into the operations database to add another level of validation for correct documentation. Complication data were collected from the clinic’s weekly internal complication conference, where all procedures performed during the last week were discussed. Postoperative complications were instantly recorded in a standardized paper-based questionnaire and later manually transferred by the neurosurgical secretariat into a digital, proprietary, relational complications database. To minimize errors, the digital layout was identical to the paper-based questionnaire. The complication database and the operation database were linked by a digital interface, avoiding data redundance (Figure 1).

### 2.2. Patient Grouping

The approach to grouping the identified spinal surgeries was to develop a standardized methodology that could be transferred to other clinics and could be compared to the available literature to evaluate the results. In the first step, the data were automatically grouped to obtain only spinal interventions as a result (Figure 1). This resulted in a main group of 5575 patients. In the second step, this group was divided into two main groups: “Primary Spinal Operations” (n = 4814) and “Revisions” (n = 761). Then, the main group was further divided by the localization (cervical (n = 1025), thoracic (n = 467), and lumbar (n = 3183)). The location-based subgroups were divided by the diagnosis (a herniated disk, degenerative condition, tumor, trauma, inflammation, and others), by the treatment type (Figure 2), and by instrumentation (yes/no). For cervical treatments, the subgroups dens surgeries (ventral screw, posterior spondylodesis) and cervical ventral fusions (cage, cage with ventral plate stabilization) were created. In addition, an extended grouping approach not affected by the localization was included. For this, the variables of age, age groups (<10, 10–19, 20–29, 30–39, 40–49, 50–59, 60–69, 70–79, 80–89, >90), gender (m/w), the surgical approach (ventral, dorsal, combined/360°), the number of previous surgeries (1, 2, 3, >3), the number of pre-existing medical conditions (1, 2, 3, >3), the ASA score (1, 2, 3, 4, 5, 6), and comorbidities were chosen.

### 2.3. Complication Definition

Complications were generally categorized as a binary event complication (yes/no). In addition, we performed a general descriptive analysis of the type of complications sorted into the groups KI (unexpected and clearly surgery-associated), KII (predictable, disease- or location-specific), and KIII/IV (non-surgical or requiring intensive care) [5]. Groups KIII and KIV were merged due to low case numbers.

### 2.4. Statistical Evaluation

The statistical analysis, performed on the pseudonymized data from the operations and complications databases, comprised descriptive analysis, hypothesis analysis, and logistic regression analysis. Bivariate hypothesis testing used the Kolmogorov–Smirnov test, *t*-test, and Mann–Whitney-U test to detect differences in the event occurrence of a complication between the defined groups/variables. Categorical variables were evaluated with Pearson’s Chi-square and Fisher’s exact test, and for multiple independent samples we used the Kruskal–Wallis test and Dunn–Bonferroni post hoc test. The identified groups with differences in the complication occurrences were further analyzed with logistic regression. In this step, we evaluated the association of the defined variables with “complication (yes)” as an outcome. A significance level of α = 5% was applied with a Bonferroni-adjusted *p*-value of ≤0.005 for multiple univariate LR tests. For the multivariate LR testing of pre-existing medical conditions, *p* < 0.05 was accepted as significant.

### 2.5. Literature Analysis

Following the receipt of the statistical results, we conducted a literature review by identifying congruences with existing studies.

## 3. Results

### 3.1. Overall Cohort Characteristics, Complications, and Mortality

Out of 5575 data records, 2921 (52.39%) were male patients and 2654 (47.61%) were female, with an average age of 57.93 ± 16.81 years for men and 59.72 ± 17.27 years for women. The average patient age increased by 5.88 years from 2006 to 2019 (*p* < 0.001), with half of the observed patients being older than 60 years. Grouped by the localization, the average age of the primary surgery group was 57.37 years for the cervical, 59.43 years for the thoracic, and 59.21 years for the lumbar spine. On average, there were 426 spinal surgeries per year, with no clear trend for more or fewer surgeries over time. The most common diagnoses were disk herniation (37.96%) and degenerative spinal diseases (36.31%), followed by trauma (14.46%), tumors (6.82%), infections (2.58%), and unclassified (1.87%). The most frequently performed operations were a sequester-/nucleotomy (31.93%), posterior spondylodesis (19.25%), and ventral fusion (10.48%), microsurgical decompression (9.78%), hemi-/laminectomies (7.68%), and foraminotomies (7.66%).

Overall, 610 (10.9%) patients had a complication, with 317 (10.85%) males and 293 (11.04%) females. The average age was significantly higher for patients with complications (63.19 vs. 58.24 years, *p* < 0.001). No statistically significant association between sex and a complication occurrence was found, regardless of the therapy. The highest complication rate (CR) was found in children under 10 years old at 19.35%, while the lowest was among adolescents and young adults aged 10–19 years at 1.33%. CRs increased with age, peaking at 16.67% in the over 90-year-old group. The complication incidence was higher in patients aged ≥50 years compared to <50 years, excluding the <10 years old group. The highest yearly CR was recorded in 2007 at 13.04%, and the lowest in 2013 at 6.90%. Nohara et al. [6] reported a similar rate of 8.6%, while Nasser et al. [7] found a combined CR of 16.4%. The results align well with these studies, with a cervical CR of 15.12%, a thoracic of 16.70%, and a lumbar of 8.55%. Additionally, the study correlates with Schoenfeld et al. (10%) [8], Sclafani et al. (11%) [9], and Kimmell et al. (9.9%) [10]. Furthermore, Sarnthein et al. [11] recorded a CR of 20% for any adverse event in a similar monocentric study design, supporting the concept of patient registries within departments for quality improvement. Of the types of complication registered in complication category KI, implant misplacement (n = 72) was the most common complication, followed by rebleeding (n = 51) and “other” complications (n = 48). Peripheral nerve lesions and cerebrospinal fistulas each accounted for n = 41. In category KII, early recurrence (n = 34) and implant failure (n = 33) were most common. For KIII/IV, “other” complications (21.05%) had the lead, followed by cardiac complications (18.95%, n = 18), pneumonias (11.58%, n = 11), thromboses (8.42%, n = 8), and pulmonary embolisms (7.37%, n = 7).

A total of 14 deaths (0.25%) were recorded at the complications conferences. Out of these, 11 (0.19%) were a direct result of complications. Three patients died due to the severity of their underlying conditions unrelated to peri- or postoperative complications. The causes of death were distributed among the complication groups as follows: Group KI included septic wound healing disorders (n = 1), implant misplacement (n = 1), and bleeding (n = 1). Group KII included perioperative media infarction with brain ischemia (n = 1). Groups KIII/IV included fulminant pulmonary embolism (n = 1), cardiac complications (n = 3), pulmonary sepsis (n = 2), sepsis with disseminated intravascular coagulation (n = 1), and one case with an unclear cause. In terms of mortality, Wang et al. [12] registered 0.14%. Smith et al.’s [13] retrospective review of mortality rates in spinal surgeries was allied with our findings, showing increasing mortality rates by age (50–59 = 0.2%; 80–89 = 0.64%), the ASA score (I = 0%; II = 0.12%; III = 0.42%; IV = 4.17%), and instrumentation (yes = 0.39%; no = 0.09%). Similarly, Goz et al. [14] reported mortality rates for spinal fusion operations that matched our findings, with overall mortality rates of 0.35% for the cervical, 0.4% for the thoracic, and 0.13% for the lumbar spine. 

### 3.2. Identified Associations with Complications

The conducted hypothesis tests investigated the defined variables for statistical significance in association with the event of a complication. Among the variables examined, age, the localization, the diagnosis group, the therapy group, instrumentation, the ASA classification, the surgical approach, pre-existing conditions, and the number of pre-existing conditions were found to be statistically significant (*p* < 0.05) in terms of their association with the occurrence of a postoperative complication. The variable “Revision” was not significantly associated with the occurrence of complications, with a *p*-value of 0.571, as was gender (*p* = 0.823). Table 1 shows the results of the logistic regression analysis with the identified statistically significant associations with the occurrence of complications.

### 3.3. Age

Age significantly influenced the likelihood of complications, as shown in Table 2, with a 2% increased risk per year (odds ratio, OR = 1.02; 95% CI 1.01–1.02; *p* < 0.001). Comparatively, people aged 60–69 had a significantly higher CR (14.02%, n = 156; OR = 3.08, 95% CI 1.64–5.78; *p* < 0.001) than young adults aged 20–29 (5.02%, n = 11). The ages of 70–79 and 80–89 demonstrated significantly higher CRs compared to young adults, with *p*-values of 0.002 and <0.001 and ORs of 2.75 (95% CI 1.47–5.15) and 3.51 (95% CI 1.82–6.74), respectively. Thus, patients between 60 and 89 had approximately three times the risk of complications (reference group aged 20–29 years.). The group <9 years old had the highest CR (19.35%, n = 6), albeit not a statistically significant one (*p* = 0.006). In the literature search, age was found to significantly influence the probability of complications. Patient age, especially an age of >65 years, is recognized as having an association with spinal surgery complications [8,10,12,15,16,17,18,19,20]. Although patients got older, the mortality rates in the presented study cohort and in the literature remained relatively constant [14].

### 3.4. Localization

Few studies consider the surgical location as having an association with complications. The logistic regression revealed a significant link between the surgical site location and CRs, with 15.12% (n = 155) for the cervical, 16.7% (n = 78) for the thoracic, and 8.55% (n = 272) for the lumbar spine. Relative to the lumbar spine, the cervical spine had an OR of 1.91 (95% CI 1.54–2.36, *p* < 0.001), while the thoracic spine showed an OR of 2.15 (95% CI 1.63–2.82, *p* < 0.001). According to our evaluation, a cervical and thoracic location compared to lumbar localization was significantly associated with complications. Cervical and thoracic patients registered higher numbers of instrumentations, tumors, and traumas compared to lumbar patients, the complications of which mainly comprised degenerative spinal diseases and lumbar disk herniations. An analysis revealed that cervical spine patients were significantly younger on average. The literature showed earlier symptomatic occurrences of cervical disk herniations (fourth decade of life) compared to lumbar herniations (fifth decade) [21,22]. Goz et al. [14] reported thoracic fusions hold a higher risk than cervical and lumbar ones. Lee et al. [17] classified complications into six categories based on the affected organ system and divided surgical locations into cervical, thoracic, lumbar, and sacral. Univariant analysis showed higher ORs for thoracic surgeries for pulmonary, neurological, hematological, and urological complications, but multivariate analysis retained only increased surgical invasiveness and an age of >65 as significant risk factors.

In the cervical diagnosis subgroup (n = 1025), cervical disk herniation (n = 28, 9.59%) was defined as the reference group. The tumor group (n = 17, 33.33%; OR of 4.71, 95% CI 2.34–9.5; *p* < 0.001), trauma group (n = 45, 19.65%; OR of 2.31, 95% CI 1.39–3.83; *p* = 0.001), and “other diagnoses” group (n = 5, 41.67%; OR of 6.73, 95% CI 2–22.63; *p* = 0.002) showed significantly increased CRs. The ‘other diagnoses’ group consisted of a very small number of cases (n = 12), including patients with an epidural or intraspinal hematoma (n = 5), syringomyelia (n = 2), cavernoma (n = 1), osteolysis (n = 1), dura fistula (n = 1), spinal cord injury (n =1), and undefined cervical vertebrae processes (n =1). CRs for degenerative spinal diseases (n = 58; 13.33%) and infections (n = 2; 33.33%) were not significant (*p* > 0.005). Patients with cervical disk herniations were significantly younger, as was also found in the study by Hammer et al. [21]. Traumas to the cervical spine showed the greatest age variance, indicating a diverse collective. Tumors and traumas showed significantly higher complication probabilities compared to cervical disk herniations. Lee et al. [20] identified tumors and traumas as having a significant association with hematological, pulmonary, and urological complications.

Among 924 treated cervical spine patients, complications occurred in 143 (15.48%). Compared to the reference group of ventral fusion, which was the most frequent (n = 547; CR of 12.43%), a hemi-/laminectomy had a significantly higher complication risk (n = 34, CR of 41.18%; OR = 4.93; 95% CI 2.38–10.22; *p* < 0.001), as did vertebral body replacement (n = 63, CR 26.98%; OR = 2.6, 95% CI 1.41–4.8; *p* = 0.002). Dens surgeries (n = 45; CR of 22.22%) and 360° fusions (n = 41; CR of 26.83%) had higher CRs, but these were not statistically significant. About two-thirds of complicated cervical vertebral body replacements were for diagnoses of tumors or fractures. About a quarter of these complications were attributed to implant misplacements or failures. A prospective case study by Arts et al. [23] reported perioperative complications in 30% of patients undergoing expandable vertebral body replacement procedures, with implant dislocation also being the most common.

Overall, 3183 patients were included in the lumbar diagnosis group, of which n = 272 (8.55%) had complications. Lumbar disk herniations (n = 94; 6.24%) were the reference group. Compared to them, patients with degenerative spinal diseases in the lumbar spine (n = 125; 10.03%) had a significantly higher risk of complications (OR 1.68; 95% CI 1.27–2.22; *p* < 0.001). Similarly, patients with tumors (n = 14, 14.43%; OR of 2.54, 95% CI 1.39–4.64; *p* = 0.003), trauma (n = 30, 11.03%; OR of 1.86, 95% CI 1.21–2.87; *p* = 0.005), and infections (n = 9, 16.98%; OR of 3.07, 95% CI 1.46–6.49; *p* = 0.003) had increased risks.

Patients receiving treatment with lumbar disk herniations were about 3.5 years older than those with cervical disk herniations, consistent with Skaf et al. [22]. The oldest cohort was the group with degenerative spinal diseases, which was expected as they often occur in older age due to age-related diseases like osteoporosis, spinal stenosis, or spondylolisthesis. Patients with infections were significantly older. The literature showed an increased susceptibility of the immune system in aged patients [24,25]. Significantly higher complication risks were found for degenerative diseases, tumors, traumas, and infections of the lumbar spine, supported by Lee et al. [26], Uakritdathikarn [27], and Nohara et al. [6].

Out of 3139 treated lumbar patients, 270 (8.55%) experienced complications. Patients who underwent a sequestrectomy/nucleotomy (the reference group), the largest (n = 1462) and youngest group (51.66 ± 15.77 years), had a complication rate of 5.95%. Statistically significant lumbar complication rates (CRs) were found for a hemi-/laminectomy (n = 267, CR of 11.24%; OR = 2, 95% CI 1.29–3.1; *p* = 0.002), dorsal internal posterior spondylodesis (n = 507, CR of 15.19%; OR = 2.83, 95% CI 2.04–3.92; *p* < 0.001) and 360° fusion (n = 30, CR of 23.33%; OR = 4.81, 95% CI 2.01–11.52; *p* < 0.001). CRs for microsurgical decompression (9.7%) and vertebral body replacement (12.24%) were not statistically significant, similarly to those foraminotomy, vertebroplasty, and kyphoplasty. For minimally invasive transforaminal lumbar interbody fusion (TLIF), Wong et al. [28] reported 15.6% perioperative, 10.3% surgical, and 7.2% medical complications. Multisegmented fusion significantly increased the complication risk. The high complication rate of 360° fusions is due to their invasiveness and use in cases like tumors, infections, or traumas. Our data showed a nearly five times higher likelihood of complications in lumbar 360° fusions compared to the reference group. Grob et al. [29] reported CRs of 18% for single-stage and 38% for double-stage combined fusions. For comparison, the average age of our cohort was significantly older than Grob et al.’s study (65.7 years vs. 33.4 years). A study by Rumalla et al. [30] on spinal fusions in pediatric spondylolisthesis identified 360° fusion (OR of 2.41; *p* = 0.001) as having a significant association with complications, which is consistent with our findings, although their study focused only on pediatric patients. In a recent study by Ayling et al. [31], the CSORN (Canadian Spine Outcomes and Research Network) prospective national database reported a 2.4% major complication rate in lumbar spine surgery, with minor complications varying widely between institutions (7.9%–42.5%).

Patients undergoing cement augmentation procedures, vertebroplasty (70.33 ± 11.18 years), and kyphoplasty (76.5 ± 5.73 years), generally the oldest patients, showed no statistically significant CR (4.35%) in comparison.

### 3.5. Instrumentation

We examined 4814 cases involving surgical instrumentation, comprising 1025 cervical, 467 thoracic, and 3183 lumbar procedures. The utilization of instrumentation rose from 23.2% in 2006 to 47% in 2018, with an overall rate of increase of 38.4% between 2006 and 2019. Notably, instrumentation was most prevalent in the cervical spine (85.56%), followed by the thoracic (62.31%) and lumbar (19.76%) regions. Older patients exhibited a higher likelihood of instrumentation in the cervical and lumbar areas. The analysis of lumbar spine instrumentation was associated with higher complication rates (CR of 15.1%, OR of 2.39, 95% CI 1.83–3.12, *p* < 0.001) compared to non-instrumented lumbar cases (CR of 6.93%). Conversely, no such association was observed for cervical and thoracic procedures. The findings, along with the existing literature, reveal a consistent upward trend in the use of instrumentation across all spinal regions. Over the study period, the proportion of instrumented surgeries nearly doubled from 23.2% in 2006 to 47.0% in 2018. A 2005 study by Deyo et al. [32] noted an increase in lumbar fusion surgeries in the US during the 1990s due to the development of new surgical instruments and techniques. Instrumentation significantly influenced the likelihood of complications in lumbar surgeries, with a 2.5-fold increased risk. After excluding the large group who underwent lumbar sequestrectomy/nucleotomy, the risk was still doubled for instrumented cases. Previous research, including Deyo et al., 2010 [33], and Kimmell et al., 2015 [10], reported increased risks and costs with spinal fusions and invasive procedures. Yadla et al. [34] reported significant risks for adult scoliosis surgeries with a 13% likelihood of pseudarthrosis and a perioperative complication incidence of more than 40%. Campbell et al., 2011 [35], found a 56% complication rate with instrumentation, influenced by the number of fused spinal segments and comorbidities like diabetes or heart disease.

### 3.6. ASA

Among the 3142 patients with recorded ASA scores, as seen in Table 3, the distribution was as follows: an ASA of 1 (12%, n = 377), an ASA of 2 (55.41%, n = 1741), an ASA of 3 (31.7%, n = 996), and an ASA of 4 (0.89%, n = 28). Elevated ASA scores were associated with increased complication rates (CRs). Specifically, we observed complication rates of 6.9% (n = 26) for an ASA of 1, 9.42% (n = 164) for an ASA of 2, 16.47% (n = 164) for an ASA of 3, and 32.14% (n = 9) for an ASA of 4. Statistical analysis revealed that an ASA of 3 and an ASA of 4 were significantly associated with higher complication rates, with odds ratios (ORs) of 2.66 (95% CI 1.73–4.1; *p* < 0.001) and 6.39 (95% CI 2.63–15.53; *p* < 0.001), respectively. However, no statistically significant association was found for an ASA of 2.

The results are reinforced by the reports described in the literature by Sobottke et al., 2012 [16], Schoenfeld et al., 2013 [8], and Kimmell et al., 2015 [10], that have repeatedly identified the ASA score as having a significant association with complications. Whitmore et al. [36] showed that both the Charlson Comorbidity Index (CCI) and the ASA score were significantly associated with complications, and a higher ASA score was also correlated with higher treatment costs.

### 3.7. Surgical Approach

Of 5282 patients, the dorsal approach was the most frequently used surgical method (81.7%), followed by the ventral (15.34%) and combined/360° methods (2.96%). LR analysis revealed a significantly higher complication association with ventral (OR of 1.48, 95% CI 1.17–1.87; *p* = 0.001) and combined/360° (OR of 2.02, 95% CI 1.69–3.60; *p* = 0.002) approaches when compared to the dorsal approach (CR of 8.09%, n = 369). In lumbar revision surgeries, the combined/360° approach presented a complication risk that was eight times higher (CR of 46.15%, n = 6; OR = 8.39, 95% CI 2.72–25.89; *p* < 0.001) than the dorsal approach (CR of 9.4%). The case number for combined/360° methods in revision surgeries was low (n = 13). No significant difference was found based on the spinal segment. Wang et al., 2007 [12], noted a higher risk with posterior or combined fusion in the cervical spine compared to anterior decompression. However, our study found no significant association in the cervical spine. The present study was limited by a smaller sample size compared to Wang et al.’s [12] study, which examined 932,009 patients. Arts et al. [23] reported complication rates of 50% for the anterior approach, 35% for the combined approach, and 15% for the posterior approach in 60 patients receiving expandable vertebral body replacement. Memtsoudis et al., 2012 [37], found complications to be significantly more frequent in combined anterior and posterior fusions than in anterior or posterior fusions only. Campbell et al. [35] found a significant association between the combined anterior–posterior approach and an increased complication risk.

### 3.8. Pre-Existing Medical Conditions

The data presented here show that 54.7% of the patients had at least one pre-existing condition. The number of pre-existing conditions was significantly associated with postoperative complications, as shown in Table 4. Patients with two, three, and ≥four pre-existing conditions were at a higher risk of complications than those with zero. The relative risk was highest for patients with three pre-existing conditions (18.57%, OR = 2.52, 95% CI 1.91–3.31, *p* < 0.001). Patients with ≥four (15.77%, OR = 2.06, 95% CI 1.42–3.00, *p* < 0.001) and two pre-existing conditions (OR = 1.68, 95% CI: 1.32–2.13, *p* < 0.001) also had a significant association. One pre-existing condition had an increased risk compared to the reference group (10.86% vs. 8.31%), but this was not statistically significant (*p* = 0.008). Table 5 shows a multivariate regression analysis revealing the following statistically significant (*p* < 0.05) results for a higher complication association: hypertension (OR of 1.22, 95% CI 1.02–1.46; *p* = 0.034), osteoporosis (OR of 1.33, 95% CI 1.02–1.74; *p* = 0.037), arrhythmias (OR of 1.57, 95% CI 1.15–2.14, *p* = 0.005), oncological comorbidities (OR of 1.61, 95% CI 1.06–2.43; *p* = 0.025), renal dysfunction (OR of 1.69, 95% CI 1.08–2.64; *p* = 0.022), a stroke history (OR of 3.09, 95% CI 1.81–5.29; *p* < 0.001), and thromboses (OR of 2.13, 95% CI 1.08–4.18; *p* = 0.028). We could not identify a statistically higher complication risk for diabetes or coronary heart disease. Studies such as those by Campbell et al. [19], Glassman et al. [38], Kimmell et al. [10], Lee et al. [26], and Bjerke et al. [39] support our results regarding the influence of pre-existing conditions on the complication rate. The data confirm that arterial hypertension and oncological diseases are associated with increased complications. However, for heart failure or diabetes, we could not demonstrate any significance.

### 3.9. Revision Surgery

While other authors and this study found no association between revisions and complications [16,20,28,40], a retrospective study by Proietti et al., 2013 [41], with 338 patients undergoing spinal surgeries from 2007 to 2011 established a correlation between increased infection rates and revision surgeries.

## 4. Discussion

The retrospective study data presented here aimed to validate a grouping and registration methodology in a level-one spine single center by comparing the findings with the existing literature. We found associations with complications consistent with the available literature, particularly noting higher rates in older patients, cervical operations, tumor surgeries, and fusion procedures. Instrumentation merits careful consideration due to its benefits and potential risks, especially amidst increasing surgery rates. Patient-specific risk profiles, including ASA scores, should inform decision-making to optimize outcomes and minimize complications. The choice of surgical approach is often dictated by the diagnosis, but it remains a relevant factor for preoperative risk assessment, particularly for combined procedures. Understanding the implications of different approaches is crucial for patient consultation and risk assessment. Additionally, the findings stress the significance of accounting for patients’ pre-existing conditions in complication risk assessments, aligning with the existing literature. This highlights the need for a comprehensive approach to risk assessment that takes into account patient characteristics, surgical factors, and institutional differences.

While these results are by no means surprising, they underscore the robustness of our standardized complication registration. The significance of the standardized recording of postoperative abnormalities across centers cannot be overstated. Such uniformity would facilitate comparisons between centers and guide further investigation into institutional differences or specific treatments. This is particularly relevant given the variability in staffing, personnel training, and other factors that can influence complication rates.

The German Federal Ministry of Health is proposing a comprehensive hospital reform to enhance transparency and facilitate evidence-based policy-making [42]. This reform entails a transition away from the German Diagnosis-Related Groups (G-DRGs) system towards the establishment of a register where patients can choose treatment facilities based on quality measures such as the number of cases treated, available equipment, and personnel expertise. These data show that complication rates should also be factored into these quality assessments. However, obtaining current and accessible data to evaluate the quality of hospitals and facilities remains a challenge, as there are currently no national registries providing a comprehensive overview.

The path described in this study is a transferable approach, involving retrospectively analyzing standardized documented complications, which could serve as a model for other medical centers to enhance comparability. However, a significant challenge with this approach is the lack of reliable complication data. Therefore, we advocate for the implementation of complication conferences with standardized registration processes to address this issue.

The involvement of nursing staff is necessary for the comprehensive and complete recording of complications during the inpatient stay. Interdisciplinary, clearly defined, and standardized recording is therefore the core element for the appropriate presentation of complications. In the procedure described here, complications are recorded by the surgeons themselves, which can, however, lead to errors in the recording process. The fact that there are different levels of the surgical hierarchy and competing surgical thinking means that, for example, the events recorded by the surgeons may be subject to bias, which means that complications may not be fully recorded (authors’ note: there is an average proportion of unreported cases of 10% per year despite the standardized regular recording of complications: unpublished data). A potential method for a consistent standardized complication definition could be the SAVES-V2 severity system proposed by Rampersaud et al. [43,44].

### 4.1. Risk Stratification and Risk Adjustment

Risk stratification is an essential approach to enhancing healthcare quality. It involves identifying preoperative patient risk factors and integrating them into decision-making processes. This integration can help prevent complications, reduce the treatment duration, and boost patient satisfaction. An integral aspect of risk stratification is selecting relevant risk associations. It is crucial that these risk associations are clinically meaningful and reliable, adequately reflecting the risk profile of patients. Predictor selection should be based on both clinical expertise and empirical evidence. The collection of risk associations aims to contribute to this field for spinal surgeries.

Another method for considering preoperative risk factors is indicator-specific risk adjustment. This entails comparing the outcomes of medical treatments or procedures based on patient-related risks. Risk adjustment is particularly important when comparing healthcare facilities, as patient mixes may vary regarding pre-existing risk factors. Hospitals with complex cases and consequently higher complication rates could be perceived as qualitatively inferior without appropriate risk adjustment compared to hospitals with uncomplicated cases.

### 4.2. Future Practical Use of Generated Data

The four main ethical principles of medicine are: respect for patient autonomy, non-maleficence, beneficence, and justice [45]. These ethical principles serve as guiding values for healthcare professionals in patient care, aiming to ensure high-quality, morally responsible, and professional healthcare delivery. Complications play a central role in light of these principles. Avoiding complications falls under the principle of non-maleficence, which dictates that healthcare professionals should strive to prevent or minimize harm to patients. Preventing complications is crucial to ensure the safety and effectiveness of medical treatment.

Furthermore, the principle of justice is implicated in the avoidance of complications, particularly concerning the fair distribution of resources in healthcare. By accurately predicting complications, the healthcare system can save costs that would otherwise be incurred for treating complications and associated sequelae. Additionally, complications can lead to longer hospital stays, resulting in higher costs and resource constraints in patient care. Avoiding complications can help alleviate these constraints and improve the care of all patients.

To make evidence-based decisions regarding the suitability of an intervention for a patient, it is essential to have appropriate data for risk stratification. These data assist us in selecting the most suitable therapeutic approach and enable high-quality and informed patient counseling. Risks play a crucial role, especially in patient education and preparation for surgical procedures. A vital aspect of medical counseling is to collaboratively establish a decision-making framework with the patient, considering the specific circumstances of the disease and its course.

Patients should be informed about the likelihood of success, as well as the potential failures and risks of the intervention. This is referred to as “informed consent” and is a critical component of medical education [46]. Thus, it is important not only for the physician to conduct careful treatment but also to uphold the patient’s right to self-determination.

One possible approach for using the collected data could be in a model for predicting complications, as described by Kimmell et al. [10]. For this purpose, each complication association would be assigned a respective weight, and the practitioner could subsequently calculate an individual risk score using patient information. This enables both the patient and the physician to assess the risk for the respective therapy procedure. For decisions regarding elective procedures with sufficient preparation time, models utilizing logistic regression and incorporating all known patient information are potential methods. For decisions that must be made quickly, heuristic models could be appropriate. These models use only the most relevant factors for the therapy decision. If one of the factors applies, further information is ignored, and the therapy is initiated; otherwise, the next predictor is considered. This allows for a treatment decision to be made in the shortest possible time [47].

Current research includes AI-driven prediction models to support therapy decisions. Unlike logistic regression, these models operate based on pattern recognition, allowing artificial intelligence to identify specific patterns in large datasets and, for example, identify specific risk associations for complications. A retrospective study by Wang et al. [48] utilized a predictive model (multivariate logistic regression) and machine learning (an XGBoost algorithm: tree-based extreme gradient boosting) to identify patients at an increased risk for venous thromboembolism following a single-segment, posterior lumbar fusion operation due to degenerative spinal pathology. The model had a higher predictive accuracy than traditional risk stratification using the ASA and Charlson Comorbidity Index (CCI). The model identified five significant clinical variables, including an age over 65, an obesity grade II or higher, coronary artery disease, the functional status, and a prolonged operation time. The model had an AUC of 0.716, comparable to the AUC value of the XGBoost algorithm. Predictive models and AI-based machine learning could contribute to clinical decision-making and risk management in the perioperative care of patients.

Saravi et al. [49] conducted a narrative literature review in 2022 on “AI-supported prediction modeling and decision-making in spinal surgery using hybrid machine learning models.” They concluded that artificial intelligence and hybrid machine learning models can assist physicians in predicting outcomes and the likelihood of failures and identifying disease patterns in multimodal data. Integrating multimodal data into new machine learning models could provide a better representation of complex patient information. Although the algorithms used were developed with large databases, they must still be questioned and considered only as decision support. Continuous updating is necessary to incorporate new data and research into existing algorithms. The presented data are an addition to available data for future AI analysis.

In 2023, the “German Ethics Council” issued recommendations on the topic in its statement “Humans and Machines—Challenges posed by Artificial Intelligence”, particularly for the medical field [50].

### 4.3. Limitations

The strength of this study lies in the large patient sample and the real-life experience it reflects. Weaknesses include the single-center design, limited follow-up, generalizability issues, and technical challenges in patient identification for database entry. Conducting the study at a single center may have limited the generalizability of the findings to other settings. Results obtained from a single center may not represent the diversity of patient populations, clinical practices, and resources available in different healthcare settings. Factors such as patient demographics, the geographic location, and institutional practices can vary, impacting the external validity of the study. Future analyses should leverage national (DGW, German Spine Registry) or international (Spine Tango) prospective registries with standardized variables to enhance comparability and validation. While our retrospective dataset provides valuable insights, its alignment with these emerging registries remains limited due to differences in the data structure and collection methods.

The methodology of standardized registration and complication assessment through a proposed complication conference could be transferred to other centers and, through that, make data more comparable. As seen in recent comparable studies, the methodology of complication conferences is an accepted method to assess complications [51,52,53]. The study may have had a limited follow-up duration due to looking at the postoperative complications during the inhouse course, which could affect the ability to capture long-term outcomes or complications. Short follow-up periods may not provide a comprehensive understanding of the effects of interventions over time. This limitation is inherent to the study design, as the aim was to analyze inpatient complications for the improvement of immediate postoperative care. Technical challenges related to patient identification may have affected the accuracy or completeness of data entry into the database. This could have introduced bias or errors in the analysis, potentially impacting the reliability of the study findings. By using standardized processes in registration and documentation, we tried to minimize the effects. In the future, a digital patient record could eradicate this issue, although data protection regulations could interfere with scientific needs.

## 5. Conclusions

We were able to demonstrate an overview for spinal surgery-associated complications in a single center. We showed that the systematic recording of complication data is viable for retrospective analysis to find associations with the occurrence of complications. The knowledge gains of this study are to be seen in the proposal of a methodology for other centers to register abnormalities after spinal surgeries. Through that, centers and their data will become more comparable and specific differences can be further analyzed. It would be desirable to have large national or international databases for future analysis. The rapid development in the field of artificial intelligence will enable the automated recognition of individual warning patterns in patients. This in turn opens up the possibility of creating objective criteria for the integration of a personalized preoperative risk profile into patient-related decision-making processes.

## Figures and Tables

**Figure 1 jcm-14-00312-f001:**
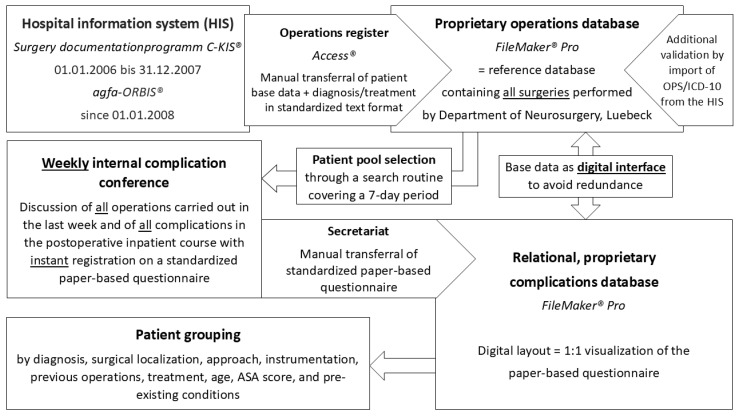
Representation of the data generation process for complication recording in the Department of Neurosurgery at University Hospital Luebeck.

**Figure 2 jcm-14-00312-f002:**
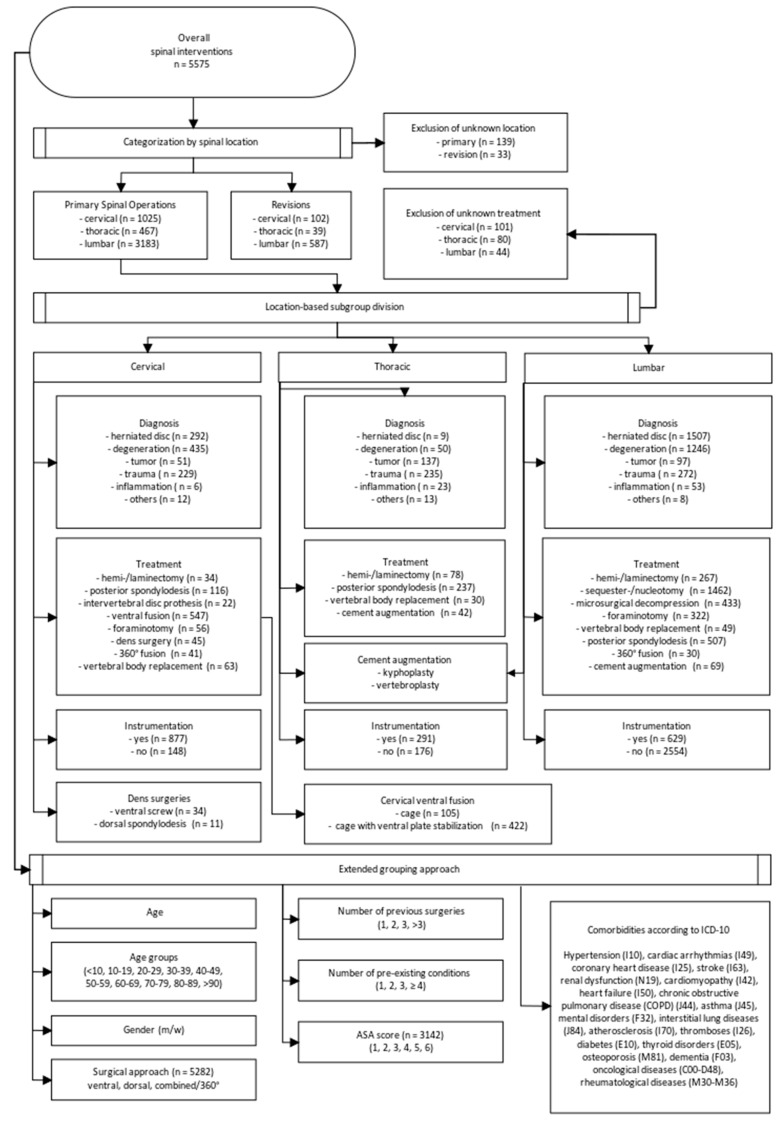
Flowchart visualizing the data grouping process.

**Table 1 jcm-14-00312-t001:** Identified statistically significant associations with spinal complications using LR.

Category	Subcategory	Subgroup	*p*	OR	95% CI
Age in years			<0.001	1.02	1.01–1.02
Age group		60–69	<0.001	3.08	1.64–5.78
		70–79	0.002	2.75	1.47–5.15
		80–89	<0.001	3.51	1.82–6.74
Localization	cervical		<0.001	1.91	1.54–2.36
	thoracic		<0.001	2.15	1.63–2.82
Diagnosis	cervical	tumor	<0.001	4.71	2.34–9.5
		trauma	0.001	2.31	1.39–3.83
	lumbar	degeneration	<0.001	1.68	1.27–2.22
		tumor	0.003	2.54	1.39–4.64
		trauma	0.005	1.86	1.21–2.87
		infection	0.003	3.07	1.46–6.49
Treatment	cervical	hemi-/laminectomy	<0.001	4.93	2.38–10.22
		vertebral body replacement	0.002	2.60	1.41–4.8
	lumbar	hemi-/laminectomy	0.002	2.00	1.29–3.1
		posterior spondylodesis	<0.001	2.83	2.04–3.92
		360° fusion	<0.001	4.81	2.01–11.52
Instrumentation	lumbar	yes	<0.001	2.39	1.83–3.12
ASA score		3	<0.001	2.66	1.73–4.1
		4	<0.001	6.39	2.63–15.53
Surgical approach	primary	ventral	0.001	1.48	1.17–1.87
		combined/360°	0.002	2.02	1.69–3.60
	revision–lumbar	combined/360°	<0.001	8.39	2.72–25.89
Pre-existing		2	<0.001	1.68	1.32–2.13
medical conditions		3	<0.001	2.52	1.91–3.31
		≥4	<0.001	2.06	1.42–3.00
	comorbidities	hypertension	0.034 *	1.22	1.02–1.46
		osteoporosis	0.037 *	1.33	1.02–1.74
		arrhythmias	0.005 *	1.57	1.15–2.14
		oncological	0.025 *	1.61	1.06–2.43
		renal dysfunction	0.022 *	1.69	1.08–2.64
		stroke history	<0.001 *	3.09	1.81–5.29
		thrombosis	0.028 *	2.13	1.08–4.18

Adjusted *p*-value significant at ≤0.005; * multivariate logistic regression significant at *p*-value of ≤0.05.

**Table 2 jcm-14-00312-t002:** Association between age and postoperative complications detected by LR.

	Average ± SD	Complication n (%)	*p*	OR	95% CI
Age in years	63.19 ± 15.54	610 (10.9%)	<0.001	1.02	1.01–1.02
Age groups in years	Overall n (%)				
<9	31 (0.56%)	6 (19.35%)	0.006	4.54	1.54–13.33
10–19	75 (1.35%)	1 (1.33%)	0.195	0.26	0.03–2.01
20–29	219 (3.93%)	11(5.02%)	Ref	-	-
30–39	474 (8.5%)	31 (6.54%)	0.438	1.32	0.65–2.68
40–49	827 (14.83%)	62 (7.50%)	0.204	1.53	0.79–2.96
50–59	1007 (18.06%)	96 (9.53%)	0.035	1.99	1.05–3.79
60–69	1113 (19.96%)	156 (14.02%)	<0.001	3.08	1.64–5.78
70–79	1331 (23.87%)	169 (12.70%)	0.002	2.75	1.47–5.15
80–89	486 (8.72%)	76 (15.64%)	<0.001	3.51	1.82–6.74
>90	12 (0.22%)	2 (16.67%)	0.111	3.78	0.74–19.40

Model age: N = 5575, R^2^ (Nagelkerkes R^2^ = 0.02; Chi2 = 48.2; df 1; *p* ≤ 0.001); model age groups: N = 5575, R^2^ (Nagelkerkes R^2^ = 0.03; Chi2 = 70.88; df 9; *p* ≤ 0.001); adjusted *p* value of ≤0.005.

**Table 3 jcm-14-00312-t003:** Association between ASA score and postoperative complications detected by LR.

	Overall n (%)	Complication n (%)	*p*	OR	95% CI
ASA Score	3142 (100%)	363 (11.55%)			
1	377 (12.0%)	26 (6.9%)	Ref	-	-
2	1741 (55.41%)	164 (9.42%)	0.122	1.4	0.91–2.16
3	996 (31.7%)	164 (16.47%)	<0.001	2.66	1.73–4.10
4	28 (0.89%)	9 (32.14%)	<0.001	6.39	2.63–15.53

N = 3142, R^2^ (Nagelkerkes R^2^ = 0.03; Chi2 = 46.88; df 3; *p* ≤ 0.001), adjusted *p*-value of ≤0.005.

**Table 4 jcm-14-00312-t004:** Association between number of pre-existing medical conditions and postoperative complications detected by LR.

Pre-Existing- Condition	Overall n (%)	Complication n (%)	*p*	OR	95% CI
Number	5575 (100%)	610 (10.94%)			
0	2526 (45.31%)	210 (8.31%)	Ref	-	-
1	1437 (25.78%)	156 (10.86%)	0.008	1.34	1.08–1.67
2	908 (16.29%)	120 (13.22%)	<0.001	1.68	1.32–2.13
3	463 (8.3%)	86 (18.57%)	<0.001	2.52	1.91–3.31
≥4	241 (4.32%)	38 (15.77%)	<0.001	2.06	1.42–3.00

N = 5575, R^2^ (Nagelkerkes R^2^ = 0.02; Chi2 = 52.52; df 4; *p* ≤ 0.001), adjusted *p*-value of ≤ 0.005.

**Table 5 jcm-14-00312-t005:** Association between pre-existing medical condition and postoperative complications detected by LR.

	Overall n (%)	Complication n (%)	*p*	OR	95% CI
Pre- Existing- Condition	5575 (100%)	610 (10,94%)			
Art. hypertension	2284 (40.97%)	296 (12.96%)	0.034	1.22	1.02–1.46
Diabetes mellitus	749 (13.43%)	102 (13.62%)	0.406	1.11	0.87–1.41
Coronary heart disease	506 (9.08%)	70 (13.83%)	0.660	1.07	0.80–1.42
Osteoporosis	487 (8.74%)	73 (14.95%)	0.037	1.33	1.02–1.74
Thyroid disorder	375 (6.73%)	56 (14.93%)	0.147	1.25	0.92–1.70
Arrythmia	318 (5.70%)	60 (18.87%)	0.005	1.57	1.15–2.14
Oncologic condition	169 (3.03%)	29 (17.16%)	0.025	1.61	1.06–2.43
Renal dysfunction	133 (2.39%)	27 (20.3%)	0.022	1.69	1.08–2.64
Stroke in history	68 (1.22%)	20 (29.41%)	<0.001	3.09	1.81–5.29
Thrombosis	48 (0.86%)	12 (25.0%)	0.028	2.13	1.08–4.18

N = 5575, R^2^ (Nagelkerkes R^2^ = 0.02; Chi2 = 67.65; df 10; *p* ≤ 0.001), *p*-value of ≤0.05.

## Data Availability

The data that support the findings of this study are available from the corresponding author upon reasonable request.

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
