# Peer review of "Retrospective Single-Center Analysis of 5575 Spinal Surgeries for Complication Associations and Potential Future Use of Generated Data"

_jcm, 2025, doi:10.3390/jcm14020312_

Round 1
Reviewer 1 Report
Comments and Suggestions for Authors
This study analyzes 5575 spinal surgeries to identify complication associations using a standardized, single-center methodology, emphasizing potential applications for AI-driven risk stratification. Here are my comments;
1. No control over confounding variables or patient follow-up beyond hospital stays.
2. Although some comparisons with literature are made, the absence of national or international registry data weakens the validation of findings.
3. Surgeon-reported complications may underreport due to bias or oversight (e.g., 10% unreported cases per year).
4. Focuses on immediate postoperative complications during hospitalization, neglecting long-term outcomes or delayed complications.
5. While AI potential is discussed, no specific AI-driven analyses or risk models are developed within the study.
6. Lack of control for surgical skill variability, institutional factors, or evolving surgical techniques over the 13-year span.
7. Data unavailability due to ethical constraints limits reproducibility and external validation.
Author Response
Comment 1: No control over confounding variables or patient follow-up beyond hospital stays.
We acknowledge the reviewer’s concern regarding the lack of control over confounding variables and follow-up beyond the hospital stay. As highlighted in the manuscript, the primary aim of this study was to document and analyze complications during the inpatient period to optimize local clinical workflows and immediate postoperative care. Long-term complications were beyond the scope of this work, as the study was not designed for follow-up data collection. We have clarified this limitation in the manuscript and highlighted the focus on hospital-based outcomes, which is a valid research method in complication analysis as seen in other similar studies (Comparison to other studies can be seen in Table 4 of Dao Trong et. al study: “Dao Trong, P., Olivares, A., El Damaty, A. et al. Adverse events in neurosurgery: a comprehensive single-center analysis of a prospectively compiled database. Acta Neurochir 165, 585–593 (2023). https://doi.org/10.1007/s00701-022-05462-w“
__________________________________________________________________________________
Comment 2. Although some comparisons with literature are made, the absence of national or international registry data weakens the validation of findings.
We agree with the reviewer that the absence of registry data limits broader validation. However, national registry data for spinal surgeries in our region have only been collected since 2012, whereas our dataset spans 13 years (starting from 2006). Retrospective alignment with such registries is therefore limited. We have now mentioned this constraint explicitly in the limitation section and acknowledged recent study like the CSORN prospective study by Ayling et al. and the prospective data by Dao Trong et al.
__________________________________________________________________________________
Comment 3. Surgeon-reported complications may underreport due to bias or oversight (e.g., 10% unreported cases per year).
We appreciate the reviewer’s observation about potential underreporting of surgeon-reported complications. To address this concern:
- All surgeries were systematically recorded in our internal operations database.
- Weekly complication conferences were held to discuss postoperative outcomes and document complications, which were manually recorded on paper and subsequently digitized.
- This rigorous process ensured that all complications discussed during these meetings were captured and transferred into the digital database, maintaining a 1:1 correspondence with the paper records.
- Furthermore, the database included ICD-10/OPS codes extracted from the hospital information system (HIS), providing an additional layer of verification.
We believe that these measures significantly minimized the likelihood of underreporting. However, we acknowledge that potential bias or underreporting, particularly with complex cases of complications, remains a challenge inherent to any retrospective study. We are of the opinion that the manuscript emphasizes both the strengths of our rigorous data collection process and the inherent limitations that could affect complete reporting.
__________________________________________________________________________________
Comment 4. Focuses on immediate postoperative complications during hospitalization, neglecting long-term outcomes or delayed complications.
We agree with the reviewer that the study does not address long-term or delayed complications. This limitation is inherent to the study design, as the aim was to analyze inpatient complications for the improvement of immediate postoperative care. This limitation has been explicitly mentioned in the limitations section of the manuscript.
__________________________________________________________________________________
Comment 5. While AI potential is discussed, no specific AI-driven analyses or risk models are developed within the study.
The data generated and presented here can be valuable in the future for providing a comprehensive overview, which can be utilized to feed or validate learning systems (AI) and by that enable automated risk profiling for patients. It was not goal of this study to develop our own AI-Model.
__________________________________________________________________________________
Comment 6. Lack of control for surgical skill variability, institutional factors, or evolving surgical techniques over the 13-year span.
We appreciate the reviewer’s point and would like to clarify that the study was conducted in a single, high-volume academic center with university background and a focus on brain and spine neurosurgery. 2018, our center has been designated as a Level-1 Spine Center (Nr. 3251 2018-2022), certified by the German Spine Association (DWG), ensuring adherence to the highest surgical standards. Although surgical techniques have evolved during the study period, all procedures followed current medical standards of care. We have included these details in the revised manuscript to provide context for the surgical environment.
__________________________________________________________________________________
Comment 7. Data unavailability due to ethical constraints limits reproducibility and external validation.
We acknowledge the reviewer’s concerns regarding data availability. As this study was conducted in a university hospital setting, all data were approved by the institutional ethics board with strict provisions for data protection and confidentiality. Consequently, the dataset cannot be made publicly available. We could agree on the following mention in the article and change the data availability to the following: “The data that support the findings of this study are available from the corresponding author upon reasonable request.”
Reviewer 2 Report
Comments and Suggestions for Authors
I am a spine surgeon in Japan. This is a large case study, and the presentation of the results of the study, including interpretations based on the literature review, was easy for the reader to understand and comfortable from a surgeon's point of view.
I found the discussion to be excellent, as it was based on an understanding of limitation, which is inevitable in research that analyzes medical information such as insurance reimbursement and ICDs.
In clinical research, it is often the case that the scientific incidence of the same complication differs greatly between prospective and retrospective studies. Although there are many difficult issues in practice, such as differences in judgment of whether a complication is a complication or not depending on the surgeon's opinion, and events that make us wonder whether they are procedural factors or device failures, I agree with the author's opinion that “we advocate for the implementation of complication conferences with standardized registration processes to address this issue.”
The importance of nurse involvement was also mentioned, but I was also concerned about the barriers to implementing this complication conference as an addition to routine practice at many medical facilities. (Even with the addition of actual nurses and co-medics, there are likely to be some events that will be evaluated differently.) However, I think this study is very valuable in making the public aware of these issues.
Author Response
We sincerely thank you for your thoughtful and encouraging feedback on our study. Your acknowledgment of the clarity and practicality of our findings from a surgeon’s perspective is deeply gratifying, especially coming from an experienced spine surgeon.
We are particularly grateful for your recognition of our discussion’s balance in addressing limitations and for highlighting the value of our proposed standardized complication conference approach. Your insights into the challenges of complication classification and barriers to implementing such conferences in routine practice are invaluable and will further inform our future work.
Thank you once again for your support and for sharing your expert perspective.

Round 2
Reviewer 1 Report
Comments and Suggestions for Authors
I would like to thank the authors for their efforts and changes. I think it can be published in its current form.
Comments on the Quality of English LanguageI would like to thank the authors for their efforts and changes. I think it can be published in its current form.